# Minimax-Optimal Location Estimation

**Shivam Gupta**
The University of Texas at Austin
shivamgupta@utexas.edu

**Jasper C.H. Lee**
University of Wisconsin–Madison
jasper.lee@wisc.edu

**Eric Price**
The University of Texas at Austin
ecprice@cs.utexas.edu

**Paul Valiant**
Purdue University
pvaliant@gmail.com

## Abstract

Location estimation is one of the most basic questions in parametric statistics. Suppose we have a known distribution density $f$, and we get $n$ i.i.d. samples from $f(x - \mu)$ for some unknown shift $\mu$. The task is to estimate $\mu$ to high accuracy with high probability. The maximum likelihood estimator (MLE) is known to be asymptotically optimal as $n \to \infty$, but what is possible for finite $n$? In this paper, we give two location estimators that are optimal under different criteria: 1) an estimator that has minimax-optimal estimation error subject to succeeding with probability $1 - \delta$ and 2) a confidence interval estimator which, subject to its output interval containing $\mu$ with probability at least $1 - \delta$, has the minimum expected squared interval width among all shift-invariant estimators. The latter construction can be generalized to minimizing the expectation of any loss function on the interval width.

## 1 Introduction

We revisit one of the most basic questions in parametric statistics: location estimation. Suppose there is a shift-invariant parametric model $\{f^{(\theta)}\}_{\theta \in \mathbb{R}}$, namely that $f^{(\theta)} = f(x - \theta)$ for some distribution $f$, and we get $n$ i.i.d. samples from $f^{(\mu)}$ for some unknown parameter $\mu$. The task is to estimate $\mu$ as accurately as possible, succeeding with probability at least $1 - \delta$. This problem includes Gaussian mean estimation as a special case. Compared to general mean estimation, where we know nothing about the distribution beyond mild moment assumptions, here we are given the shape of the distribution, and only the shift is unknown. This allows us to aim for better performance than in mean estimation, and even handle cases where the mean of the distribution does not exist.

The problem has been widely studied as a special case of parametric estimation. In particular, the classic asymptotic theory [Vaa00] recommends using the maximum likelihood estimator (MLE), which, when we fix a distribution $f = f^{(0)}$ and take the number of samples $n$ to $\infty$, satisfies $\mu_{\mathrm{MLE}} - \mu \to \mathcal{N}(0, 1/(n\mathcal{I}))$ for any parameter $\mu$. Here, the variance of the asymptotic Gaussian is $1/(n\mathcal{I})$, where $\mathcal{I}$ is the *Fisher information* of the distribution $f$, defined by $\mathcal{I} \doteq \int_{\mathbb{R}} (f'(x))^2/f(x)\,\mathrm{d}x$. It is a standard fact that $1/\mathcal{I} \le \sigma^2$ for any variance-$\sigma^2$ distribution, meaning that the asymptotic performance of the MLE is always at least as good as the sample mean. Conversely, the well-known Cramér-Rao theorem states that, for any shift-equivariant estimator $\hat{\mu}$, its variance must be at least $1/(n\mathcal{I})$ as well.

However, the classic asymptotic guarantee can be misleading, since the convergence to the asymptotic Gaussian may be arbitrarily slow in the number of samples, depending on the underlying distribution $f$. Indeed, recent work by Gupta et al. [GLPV22] showed that the issue is in fact information-

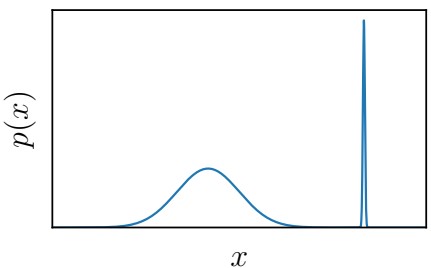

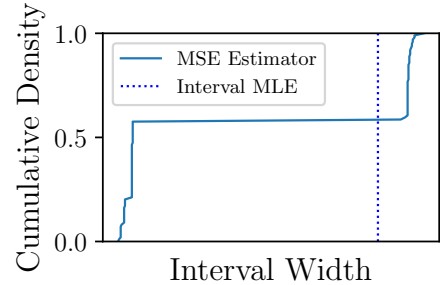

(a) The input distribution has a rare but very informative spike

(b) The minimax interval estimator from 10 samples finds the smallest possible worst-case interval, but at the cost of much worse typical interval size.

Figure 1: Example comparing our algorithms

theoretic, that no estimator can converge to the Fisher information rate in a distribution-independent manner as $n \to \infty$. They [GLPV22; GLP23] also studied the finite sample *algorithmic* theory of location estimation, with the goal of yielding finite-sample estimation error tight to within a $1 + o(1)$ multiplicative factor. To circumvent the impossibility of converging to the Fisher information in a distribution-independent finite-sample manner, they instead proposed a theory based on the *smoothed* Fisher information $\mathcal{I}_r$. That is, the Fisher information of the distribution $f$ after convolving with $\mathcal{N}(0, r^2)$. They showed that applying the MLE to perturbed samples and the smoothed distribution will yield an estimation error that converges to $\mathcal{N}(0, 1/(n\mathcal{I}_r))$ in a distribution-independent fashion, for some smoothing radius $r$ that vanishes as $n \to \infty$.

The key issue with the above theory, however, is that the Fisher information can be extremely sensitive to smoothing. As a simple example, a Dirac-$\delta$ spike has infinite Fisher information, yet smoothing it by $\mathcal{N}(0, r^2)$ reduces its Fisher information to $1/r^2$. Thus, to get $1 + o(1)$ tightness in estimation error, one would also need to also completely optimize away the smoothing. This is highly challenging and not yet achieved by these prior works on a distribution-by-distribution basis.

In this work, we instead directly construct minimax-optimal location estimators, under two different criteria. The first construction is a point estimator $\hat{\mu}$ which, for a given failure probability $\delta^*$, attains the minimum estimation error $\epsilon$ such that $|\hat{\mu} - \mu| \le \epsilon$ with probability at least $1 - \delta^*$. The second construction is a confidence interval estimator $[\mu_\ell, \mu_r]$. Consider a loss function on the size of the output interval $L(\mu_r - \mu_\ell)$ that is monotonically increasing. The second estimator is one such that, subject to the "correctness" constraint that the true parameter $\mu$ lies in the output interval $[\mu_\ell, \mu_r]$ with probability at least $1 - \delta^*$, the estimator minimizes the mean loss.

The distinction is whether we want to minimize the *worst-case* or the *average* confidence interval. For example, consider the distribution $(1 - 1/n)N(\theta, 1) + (1/n)N(\theta + n, r^2)$ for $r = \frac{1}{n^2}$ (see Figure 1). There is about a $1/e$ chance that you don't sample the narrow spike, in which case your uncertainty is effectively the Gaussian $N(0, \frac{1}{n})$; but if you do sample the spike $k > 0$ times, it gives a much more precise $N(0, \frac{r^2}{k})$ uncertainty. That is, some observed samples $x$ could give very precise estimates, while other observed $x$ cannot. The second algorithm gives worse worst-case behavior in exchange for much better average case behavior.

Our first algorithm only cares about the maximum size confidence interval it returns. That will be dominated by the $1/e$ chance the narrow spike isn't sampled; so to get a 95% confidence interval, it will always return essentially a $(1 - \frac{e}{20})$-confidence interval for the $N(0, \frac{1}{n})$ error, which is $[-\frac{1.49}{\sqrt{n}}, \frac{1.49}{\sqrt{n}}]$. Our second algorithm could instead think: by returning the slightly larger interval $[-\frac{1.50}{\sqrt{n}}, \frac{1.50}{\sqrt{n}}]$ in the cases where the narrow spike is not sampled, it can return *much* smaller $\Theta(r)$ size intervals when the spike is seen. It will optimize this tradeoff according to the given loss function; for square loss, this will be with $\Theta(r\sqrt{\log n})$ in the (more common!) "good" case, and a $1 + 1/\text{poly}(n)$ factor worse than optimal interval in the "bad" case.

## 1.1 Our results and techniques

**Minimax-optimal point estimation** In the first setting we consider in this paper: Given a point estimator $\hat{\mu}$, for a given failure probability $\delta^*$, we measure its estimation error $\epsilon_{\hat{\mu},\delta^*}$ by its worst-case error over parameters $\mu$, namely

$$\inf\left\{\epsilon > 0 \;\middle|\; \sup_{\mu} \Pr_{x_1,\ldots,x_n \sim f^{(\mu)}} \left(|\hat{\mu}(x_1,\ldots,x_n) - \mu| \leq \epsilon\right) \geq 1 - \delta^*\right\}$$

Our first main contribution is that the estimator defined in Algorithm 2 achieves (essentially) minimal estimation error.

**Theorem 1.1.** *For any small $\kappa > 0$, the estimator defined in Algorithm 2 has estimation error within a $(1 + \kappa)$ factor of minimax optimal.*

In order to construct this estimator, we solve the dual problem: fixing an estimation accuracy $\epsilon$, find an estimator that minimizes the failure probability $\delta$. The optimal estimator for this dual problem can be defined as the limit of a family of Bayes-optimal estimators. Specifically, fixing estimation error $\epsilon$, the estimator $\mathcal{A}_t$ defined in Algorithm 1 attains the minimum failure probability when the true parameter $\mu$ is drawn from the prior $\mathrm{Unif}[-t, t]$. The following theorem shows that the estimator in the limit as $t \to \infty$, $\mathcal{A}_\infty$, attains the minimax-optimal failure probability.

**Theorem 1.2.** *The algorithm $\mathcal{A}_\infty$ (limit of Algorithm 1 as $t \to \infty$) is minimax-optimal with respect to the 0-1 loss $L(\hat{\mu}, \mu) = \mathbb{1}[|\hat{\mu} - \mu| > \epsilon]$, namely, with respect to the failure probability.*

The estimator $\mathcal{A}_\infty$ induces a function of the failure probability $\delta$ in terms of the estimation error $\epsilon$. Thus, given a desired a failure probability $\delta^*$, the $\delta(\epsilon)$ function can be inverted to define the optimal estimator in Algorithm 2 in Section 3.2. Since $\delta(\epsilon)$ is monotonic, we can use binary search to approximately compute the optimal $\epsilon$. The binary search is explicitly given and analyzed in Appendix A.1.

**Minimax-optimal confidence-interval estimation** The second setting we consider in this paper concerns confidence-interval estimators. Suppose we are given an increasing loss function $L : \mathbb{R}_+ \to \mathbb{R}_+$ mapping an interval width to a loss, and given an estimator $\mathcal{A}$ whose output interval we denote by $[\mu_\ell, \mu_r]$, we measure its *mean loss* by

$$\sup_{\mu \in \mathbb{R}} \mathbb{E}_{x_1,\ldots,x_n \sim f^{(\mu)}} \left[L(\mu_r - \mu_\ell)\right]$$

Our main result for this second setting is that the estimator defined in Algorithm 4 achieves minimax-optimal mean loss, subject to the constraint that for all parameters $\mu \in \mathbb{R}$, the probability that $\mu \in [\mu_\ell, \mu_r]$ is at least $1 - \delta^*$ for a given failure probability $\delta^*$.

**Theorem 1.3.** *Consider the set $\mathrm{Alg}_{\delta^*}$ of estimators $\mathcal{A}$ whose output we denote by $[\mu_\ell, \mu_r]$, such that for all parameters $\mu$,*

$$\Pr_{x_1,\ldots,x_n \sim f^{(\mu)}} (\mu \in [\mu_\ell, \mu_r]) \geq 1 - \delta^*$$

*Consider the estimator defined in Algorithm 4 using failure probability $\delta^*$, and denote its mean loss by $R$. Then,*

$$R \leq \inf_{\mathcal{A} \in \mathrm{Alg}_{\delta^*}} \sup_{\mu \in \mathbb{R}} \mathbb{E}_{x_1,\ldots,x_n \sim f^{(\mu)}} \left[L(\mu_r - \mu_\ell)\right]$$

The construction of Algorithm 4 and proof of Theorem 1.3 come in two parts. First, we show in Section 4.1 that Algorithm 4 is minimax-optimal among *equivariant* estimators. That is, estimators which, the probability (density) of the estimator outputting the interval $[\mu_\ell, \mu_r]$ on input $(x_1,\ldots,x_n)$ is equal to the probability of the estimator outputting $[\mu_\ell + \theta, \mu_r + \theta]$ on input $(x_1 + \theta,\ldots,x_n + \theta)$. Second, we show in Section 4.2 that, for any estimator, equivariant or not, its performance can be arbitrarily well-approximated by an equivariant estimator. Together, this shows Theorem 1.3, that Algorithm 4 is minimax optimal.

The bulk of the technical work comes from the first step, constructing Algorithm 4 and showing that it is minimax-optimal among equivariant estimators. The challenge is that we need to optimize the

mean loss of an estimator *subject to* the constraint on its probability of success, which is the event that its output interval does in fact include the true parameter.

Our approach is to use convex optimization techniques: for each equivariant estimator $\mathcal{A}$, we consider its mean loss $R_{\mathcal{A}}$ and its failure probability $\delta_{\mathcal{A}}$, both of which are independent of the true parameter due to the equivariance of $\mathcal{A}$. It is easy to check that the set $\{(R_{\mathcal{A}}, \delta_{\mathcal{A}})\}$ over equivariant estimators $\mathcal{A}$ is a convex set. We give Algorithm 3, which, for any angle $\rho \in [0, \pi/2)$, is an equivariant estimator which attains the minimum possible $R_{\mathcal{A}} \cos \rho + \delta_{\mathcal{A}} \sin \rho$ among equivariant estimators. We also show the optimality of Algorithm 3 in Theorem 4.3. This estimator thus defines a failure probability $\delta_{\rho}$ and a supporting hyperplane of angle $\rho$ for the $\{(R_{\mathcal{A}}, \delta_{\mathcal{A}})\}$ set. Noting that $\delta_{\rho}$ is monotonic in $\rho$, this again means we can binary search on $\rho$ to yield the desired failure probability $\delta^*$. Appendix B.1 gives the explicit binary search algorithm and its correctness guarantees.

## 2 Related work

Location estimation, as a special case of parametric estimation, has a well-established asymptotic theory [Vaa00] concerning the MLE, whose asymptotic performance is captured by the Fisher information. There have also been works on the finite-sample performance of the MLE, see for example [Spo11; Pin17; Van00; Mia10]. However, these works impose strong regularity conditions on the parametric model, and lose at least constant multiplicative factors in their error guarantees.

More recently, Gupta et al. [GLPV22; GLP23] focused on location estimation, and aimed to characterize the finite-sample theory to within $1 + o(1)$-tightness in the estimation error. They proposed a theory in terms of the *smoothed* Fisher information. Yet, there are examples of distributions whose Fisher information decrease significantly even after little smoothing, meaning that their theory does not always capture the optimal finite-sample bounds up to a $1 + o(1)$-factor. By contrast, in this work, we directly construct minimax-optimal estimators.

The related problem of mean estimation has also seen renewed interest in its finite-sample theory, also with the aim of yielding $1 + o(1)$-tightness in the estimation error. Catoni [Cat12] showed in his seminal work that, if the variance of the underlying distribution is known, then it is possible to achieve $1 + o(1)$-tight estimation error. Catoni [Cat12] and Devroye et al. [DLLO16] also showed that, as long as the kurtosis (normalized central $4^{\text{th}}$ moment) is bounded, knowledge of the variance is unnecessary. Recently, Lee and Valiant [LV22] proposed an estimator also yielding $1 + o(1)$-tight error, but removing the knowledge assumption on the variance entirely, and instead assuming only its existence and finiteness.

## 3 Minimax-optimal point estimator

The goal of this section is to construct a point estimator $\hat{\mu}$ that yields minimax-optimal estimation error, where error for $\hat{\mu}$ is measured by $\inf \left\{ \epsilon > 0 \mid \sup_{\mu} \Pr_{x_1, \ldots, x_n \sim f^{(\mu)}} \left( |\hat{\mu}(x_1, \ldots, x_n) - \mu| \leq \epsilon \right) \geq 1 - \delta^* \right\}$.

Section 3.1 first gives and analyzes an algorithm which takes as input an estimation accuracy $\epsilon$ and returns an estimate $\hat{\mu}$ whose failure probability $\Pr(|\hat{\mu} - \mu| > \epsilon)$ is minimax-optimal. Section 3.2 then uses binary search to construct an estimator that instead takes in a desired failure probability, and outputs an estimation that (almost) optimally minimizes the estimation error.

### 3.1 Estimator with minimax failure probability

The goal of this section is to construct an algorithm that takes in a desired estimation accuracy $\epsilon$ and optimizes for the failure probability $\delta$. The key intuition is that, since we do not know what the true parameter $\theta$ is, we might as well start by constructing an estimator that is Bayes optimal with respect to the uniform prior over the real line $\text{Unif}(\mathbb{R})$. However, there is no such thing as a uniform prior over the real line. So instead, Algorithm 1 below is constructed to be Bayes optimal with respect to the prior $\text{Unif}[-t, t]$, and we take the limit of $t \to \infty$ to get a translation-equivariant estimator, whose risk (failure probability) is independent of the underlying parameter $\theta$. We then use standard tools to relate Bayes optimality to minimax optimality, under such translation equivariance.

---

**Algorithm 1** The algorithm $\mathcal{A}_t$ for a fixed estimation accuracy $\epsilon$

---

Input: Distribution shape $f$, samples $x_1, \ldots, x_n$, and estimation accuracy $\epsilon$

    1. Let $l_x(\theta)$ be the likelihood of $f^{(\theta)}$ yielding the observed samples $x_1, \ldots, x_n$.

    2. Return $\hat{\mu} = \arg\max_{\theta \in [-t+\epsilon, t-\epsilon]} \int_{\theta-\epsilon}^{\theta+\epsilon} l_x(\theta) \, d\theta$

---

As mentioned earlier, the optimal algorithm we propose is the limit $\mathcal{A}_\infty$, namely, in Step 2 of Algorithm 1, the $\arg\max$ on $\theta$ is over the entire real line. Note, as a basic observation, that $\mathcal{A}_\infty$ is an equivariant estimator.

To show that $\mathcal{A}_\infty$ is minimax optimal in failure probability, we will use the standard technique of relating it to (a sequence) of Bayes-optimal estimators for some (sequence of) Bayesian priors.

Let us define the relevant notation here.

**Definition 3.1.** *Throughout this section, we consider the 0-1 loss function $L(\mu, \hat{\mu}) = \mathbb{1}[|\hat{\mu} - \mu| > \epsilon]$ for some fixed $\epsilon$. The failure probability of an $n$-sample estimator $\hat{\mu}$ with respect to a parameter $\mu$ can then be re-expressed as the* risk $R(\mu, \hat{\mu}) = \mathbb{E}_{x_1, \ldots, x_n \sim f^{(\mu)}}[L(\mu, \hat{\mu}(x_1, \ldots, x_n))]$.

*Consider a prior $\pi$ on the parameter $\mu$ over $\mathbb{R}$. We denote the Bayes risk of an estimator $\hat{\mu}$ with respect to prior $\pi$ as $R(\pi, \mu) = \mathbb{E}_{\mu \sim \pi}[R(\mu, \hat{\mu})]$, under the usual abuse of notation.*

To show that a given estimator $\hat{\mu}$ has minimax optimal risk (namely, failure probability), the following standard fact states that it suffices to demonstrate a sequence of priors, such that the worst-case risk of $\hat{\mu}$ is the limit of the optimal Bayes risk of these priors. We prove this fact in Appendix A.

**Fact 3.2.** *Given an $n$-sample estimator $\hat{\mu}$, suppose there exists a sequence of priors $\{\pi_i\}$ such that the worst-case risk of $\hat{\mu}$ is equal to the limit of optimal Bayes risk of $\pi_i$, namely*

$$\sup_{\mu \in \mathbb{R}} R(\mu, \hat{\mu}) = \lim_{i \to \infty} \inf_{\hat{\mu}'} R(\pi_i, \hat{\mu}')$$

*Then $\hat{\mu}$ is minimax optimal.*

To show the minimax optimality of $\mathcal{A}_\infty$, we will consider the sequence of priors $\pi_t = \text{Unif}([-t, t])$. The first step is to characterize the Bayes-optimal estimators for the prior $\pi_t$.

**Lemma 3.3.** *The algorithm $\mathcal{A}_t$ is Bayes-optimal for the prior $\pi_t$.*

*Proof.* Given the samples $x_1, \ldots, x_n$, the likelihood function $l_x(\theta)$ constructed in Step 1 of $\mathcal{A}_t$, limited to the range $[-t, t]$, is equal to the posterior distribution of $\theta$ given $x$ up to a normalization factor. Thus, by construction, Step 2 finds the $\hat{\mu}$ that maximizes the posterior success probability, or equivalently, $\hat{\mu}$ that minimizes the expected posterior loss (the posterior failure probability) given $x$. Hence $\mathcal{A}_t$ is a Bayes-optimal estimator. $\qquad\square$

With Fact 3.2 and Lemma 3.3, we can now state and prove that the algorithm $\mathcal{A}_\infty$ (Algorithm 1) is minimax-optimal with respect to the 0-1 loss, namely that it has minimax-optimal failure probability.

**Theorem 1.2.** *The algorithm $\mathcal{A}_\infty$ (limit of Algorithm 1 as $t \to \infty$) is minimax-optimal with respect to the 0-1 loss $L(\hat{\mu}, \mu) = \mathbb{1}[|\hat{\mu} - \mu| > \epsilon]$, namely, with respect to the failure probability.*

*Proof.* Fact 3.2 implies that, it suffices for us to show

$$\sup_{\mu} R(\mu, \mathcal{A}_\infty) - \inf_{\hat{\mu}'} R(\pi_i, \hat{\mu}') \to 0$$

as $i \to \infty$. Since $\mathcal{A}_\infty$ is equivariant, the risk $R(\mu, A_\infty)$ is independent of $\mu$. Also, recall by Lemma 3.3 that $\mathcal{A}_t$ is Bayes-optimal for the prior $\pi_t$. The above claim is thus equivalent to

$$R(\pi_i, \mathcal{A}_\infty) - R(\pi_i, \mathcal{A}_i) \to 0$$

as $i \to \infty$.

Since $\mathcal{A}_i$ is Bayes-optimal for $\pi_i$, the left hand side is always positive. Also observe that the left hand side is upper bounded by the probability that $\mathcal{A}_\infty$ has a different output from $\mathcal{A}_i$. We will show that such probability, explicitly, $\mathrm{Pr}_{\mu \sim \pi_i; x_1, \ldots, x_n \sim f^{(\mu)}}(\mathcal{A}_\infty(x) \neq \mathcal{A}_i(x))$, tends to 0 as $i \to \infty$.

We first claim that, fixing a distribution shape $f$, for any small probability $\kappa > 0$, there exists a sufficiently large $i$ such that, if the true parameter $\theta \in [-i + \sqrt{i}, i - \sqrt{i}]$, then there is at most $\alpha/2$ probability that $\mathcal{A}_\infty$ has a different output from $\mathcal{A}_i$ when given i.i.d. samples from $f^{(\mu)}$. This is easy to see, since $\mathcal{A}_\infty$ outputting differently from $\mathcal{A}_i$ implies the output of $\mathcal{A}_\infty$ is outside of $[-i + \epsilon, i - \epsilon]$. Furthermore, since $\mathcal{A}_\infty$ is equivariant, we have

$$\Pr_{x_1, \ldots, x_n \sim f^{(\mu)}}(\mathcal{A}_\infty(x) \notin [\mu - t, \mu + t]) \to 0$$

as $t \to \infty$ for any fixed $\mu$. For $\mu \in [-i + \sqrt{i}, i - \sqrt{i}]$, the locations $-i + \epsilon$ and $i - \epsilon$ are at least $t = \Omega(\sqrt{i})$ away from $\mu$. Thus we can set $i$ sufficiently large such that $\mathcal{A}_\infty$ outputs outside of $[-i + \epsilon, i - \epsilon]$ with probability at most $\kappa/2$, for any fixed $\mu \in [-i + \sqrt{i}, i - \sqrt{i}]$.

Further observe that, under prior $\pi_i$, there is only $2/\sqrt{i}$ probability that the parameter $\mu$ is sampled be to outside of $[-i + \sqrt{i}, i - \sqrt{i}]$. By setting $i$ sufficiently large, this probability is at most $\kappa/2$, for any given $\kappa$.

In summary, for any given small probability $\kappa$, there exists a sufficiently large $i$ such that $\mathcal{A}_\infty$ outputs differently from $\mathcal{A}_i$ under prior $\pi_i$ with probability at most $\kappa$. We have thus shown that

$$R(\pi_i, \mathcal{A}_\infty) - R(\pi_i, \mathcal{A}_i) \to 0$$

as $i \to \infty$, completing the proof. $\qquad\square$

## 3.2 Estimator with (almost) optimal estimation error

In Section 3.1, we presented an algorithm which, when given the distribution shape $f$ and a fixed estimation error $\epsilon$, finds an estimate which minimizes the failure probability $\delta$. Observing that $\delta$ is a monotonic function in $\epsilon$, we can also interpret $\epsilon$ as a monotonic function of $\delta$. Thus, if we are given a fixed $\delta$ (and distribution $f$), there is an infimum over achievable $\epsilon$. Algorithm 1 therefore also induces an optimal algorithm that optimizes for $\epsilon$ when given $\delta$, as follows.

**Algorithm 2.** *Consider the optimal failure probability $\delta$ as a function of the estimation accuracy $\epsilon$. Given a desired failure probability $\delta^*$, define $\epsilon^* = \inf\{\epsilon > 0 \mid \delta(\epsilon) \leq \delta^*\}$. Using Algorithm 1 with estimation accuracy $(1 + \kappa)\epsilon^*$ for $\kappa > 0$ results in an estimator whose failure probability is by definition at most $\delta^*$. By construction, this estimator has accuracy that is within a $(1 + \kappa)$ factor of minimax optimality.*

The optimal estimation error $\epsilon^*$ can be computed using binary search. We defer the details to Appendix A.1.

# 4 Minimax-optimal confidence-interval estimator

Recall the second problem setting of this paper concerning confidence-interval estimators. Fix a loss function $L : \mathbb{R}_+ \to \mathbb{R}_+$ mapping an interval width to a loss, and assume it is increasing. Given an estimator $\mathcal{A}$ whose output interval we call $[\mu_\ell, \mu_r]$, its *mean loss* is the worst-case expected loss

$$\sup_{\mu \in \mathbb{R}} \mathbb{E}_{x_1, \ldots, x_n \sim f^{(\mu)}} [L(\mu_r - \mu_\ell)]$$

Consider the set $\mathrm{Alg}_{\delta^*}$ of estimators $\mathcal{A}$ that are "correct" with probability $1 - \delta^*$. That is, for all parameters $\mu$,

$$\Pr_{x_1, \ldots, x_n \sim f^{(\mu)}}(\mu \in [\mu_\ell, \mu_r]) \geq 1 - \delta^*$$

Then, the goal of this section is to find an estimator in $\mathrm{Alg}_{\delta^*}$ that minimizes its mean loss, for any desired failure probability $\delta^* \in (0, 1]$.

Section 4.1 shows how to define the optimal *equivariant* estimator subject to the $1 - \delta^*$ probability correctness constraint. Appendix B.1 gives a binary search procedure for approximately computing this optimal equivariant estimator. Finally, Section 4.2 shows that the optimal equivariant estimator is also optimal across all estimators.

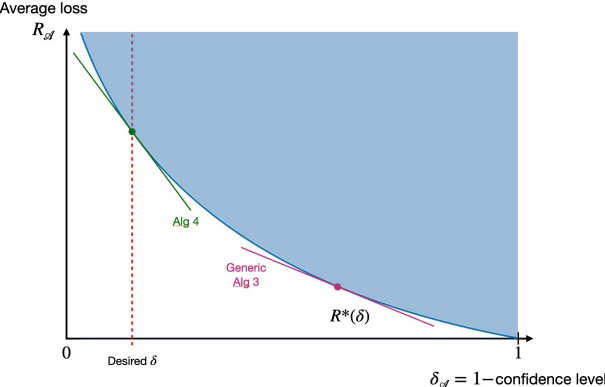

Figure 2: Illustration of the feasible set $\mathcal{F}_f$ in Definition 4.1, and Algorithm 3 and Algorithm 4

## 4.1 Optimality among equivariant estimators

We use convex optimization techniques to find the best equivariant estimator. To do so, consider the following feasible set of estimator performances, where for each equivariant estimator $\mathcal{A}$, we consider the pair $(R_\mathcal{A}, \delta_\mathcal{A})$ which is respectively its mean loss and its failure probability. This set is always convex.

**Definition 4.1.** *Fix a distribution $f = f^{(0)}$. Given a (potentially) randomized equivariant $n$-sample estimator $\mathcal{A}$ with output $[\mu_\ell, \mu_r]$, we say that $\mathcal{A}$ achieves the performance pair $(R_\mathcal{A}, \delta_\mathcal{A})$ if $\mathbb{E}_{x_1, \ldots, x_n \sim f}[(\mu_r - \mu_\ell)^2] \leq R_\mathcal{A}$ and $\Pr_{x_1, \ldots, x_n \sim f}(0 \notin [\mu_\ell, \mu_r]) \leq \delta_\mathcal{A}$.*

*Define the feasible set for distribution $f$ to be the set*

$$\mathcal{F}_f = \{(R_\mathcal{A}, \delta_\mathcal{A}) \text{ achieved by some an equivariant } n\text{-sample estimator } \mathcal{A}\}$$

*See Figure 2 for a pictorial illustration of the set.*

**Fact 4.2.** *Given an arbitrary distribution $f$, the feasible set $\mathcal{F}_f$ is convex. As a corollary, the boundary $R^*(\delta) = \inf\{R_\mathcal{A} \mid (R_\mathcal{A}, \delta) \in \mathcal{F}_f\}$ is a non-increasing convex function.*

*Proof.* Given two algorithms $\mathcal{A}_0$ and $\mathcal{A}_1$ achieving $(R_{\mathcal{A}_0}, \delta_{\mathcal{A}_0})$ and $(R_{\mathcal{A}_1}, \delta_{\mathcal{A}_1})$ respectively, consider a new randomized algorithm which runs $\mathcal{A}_0$ with probability $\lambda$ and $\mathcal{A}_1$ otherwise, for an arbitrary $\lambda \in [0, 1]$. The new algorithm achieves $(\lambda R_{\mathcal{A}_0} + (1 - \lambda) R_{\mathcal{A}_1}, \lambda \delta_{\mathcal{A}_0} + (1 - \lambda) \delta_{\mathcal{A}_1})$. Thus, $\mathcal{F}_f$ is a convex set.

Since $\mathcal{F}_f$ is a convex set, the boundary $R^*$ is also a convex function. The definition of $\mathcal{F}_f$ also directly implies that $R^*$ is non-increasing. □

Our goal then is to define the equivariant algorithm $\arg\min_\mathcal{A}\{R_\mathcal{A} \mid (R_\mathcal{A}, \delta) \in \mathcal{F}_f\}$, that in fact, the boundary $R^*$ being an inf is actually achievable. The key idea is to consider the (sub-)derivative (or just "slope", from now on) of the boundary function $R^*(\delta)$, which is always non-positive and also increasing as a function of $\delta$, since $R^*$ is non-increasing and convex. We will first show how, given a slope, we can find an estimator whose $(R_\mathcal{A}, \delta_\mathcal{A})$ pair lies on the boundary of $\mathcal{F}_f$ and has a supporting hyperplane with the desired slope (see "Generic Alg 3" in Figure 2). Having defined this estimator, its failure probability $\delta_\mathcal{A}$ can be evaluated. Since the failure probability is an increasing function of the slope, we will then do a binary search on the value of the slope to yield the correct value of $\delta_\mathcal{A} = \delta^*$ (see "Alg 4" in Figure 2).

Specifically, we will parameterize the slope by its angle $\rho \in [0, \pi/2]$. Given an angle $\rho$, we will define an estimator $\mathcal{A}$ in $\arg\min\{R_\mathcal{A} \cos \rho + \delta_\mathcal{A} \sin \rho \mid (R_\mathcal{A}, \delta_\mathcal{A}) \in \mathcal{F}_f\}$. Additionally, since the $\arg\min$ set may contain many estimators, we will also have the choice to additionally find the algorithm in the $\arg\min$ with the minimum $\delta_\mathcal{A}$ (equivalently, maximizing $R_\mathcal{A}$) or the largest $\delta_\mathcal{A}$.

As we see in the following theorem, the quantity $\|l_{\vec{x}}\|_1$ in Step 3 above is in fact the probability (density) of observing any shifted versions of $\vec{x}$, and hence exists and is finite. Thus the normalization step is well-defined.

---

**Algorithm 3** Estimator minimizing $R_\mathcal{A} \cos \rho + \delta_\mathcal{A} \sin \rho$ for a given angle $\rho \in [0, \pi/2]$

---

1. Input: samples $x_1, \ldots, x_n$, distribution $f$, Boolean flag smallestDelta indicating whether we tiebreak for the smallest $\delta_\mathcal{A}$ or the largest $\delta_\mathcal{A}$

2. Compute the likelihood function $l_{\vec{x}}(\theta)$ of $f^{(\theta)}$ yielding the observed sample $x_1, \ldots, x_n$.

3. Normalize $l_{\vec{x}}$ into the distribution $p_{\vec{x}}(\theta) = l_{\vec{x}}(\theta)/\|l_{\vec{x}}\|_1$.

4. For each interval $[\tilde{\mu}_\ell, \tilde{\mu}_r]$, define its risk $R_{[\tilde{\mu}_\ell, \tilde{\mu}_r]}$ as $(\tilde{\mu}_r - \tilde{\mu}_\ell)^2$, and its corresponding $\delta_{[\tilde{\mu}_\ell, \tilde{\mu}_r]}$ as the probability $1 - p_{\vec{x}}[\tilde{\mu}_\ell, \tilde{\mu}_r]$ under the distribution $p_{\vec{x}}$.

5. Return $[\mu_\ell, \mu_r]$ that minimizes $R_{[\mu_\ell, \mu_r]} \cos \rho + \delta_{[\mu_\ell, \mu_r]} \sin \rho$, and tiebreak according to the input Boolean flag smallestDelta.

---

The following theorem shows that Algorithm 3 is indeed an optimal estimator that minimizes $R_\mathcal{A} \cos \rho + \delta_\mathcal{A} \sin \rho$ among all equivariant estimators.

**Theorem 4.3** (Correctness of Algorithm 3). *Suppose Algorithm 3 with angle $\rho$ and flag smallestDelta set to true achieves the pair $(R_\mathcal{A}, \delta_\mathcal{A})$. Then, for an arbitrary equivariant estimator $\tilde{\mathcal{A}}$ whose performance is $(R_{\tilde{\mathcal{A}}}, \delta_{\tilde{\mathcal{A}}})$, we have that*

$$R_\mathcal{A} \cos \rho + \delta_\mathcal{A} \sin \rho \leq R_{\tilde{\mathcal{A}}} \cos \rho + \delta_{\tilde{\mathcal{A}}} \sin \rho$$

*Furthermore, if $R_\mathcal{A} \cos \rho + \delta_\mathcal{A} \sin \rho = R_{\tilde{\mathcal{A}}} \cos \rho + \delta_{\tilde{\mathcal{A}}} \sin \rho$, then $\delta_\mathcal{A} \leq \delta_{\tilde{\mathcal{A}}}$.*

*The analogous statement holds for Algorithm 3 with the flag smallestDelta set to false.*

*Moreover, as a corollary, we have $R^*(\delta_\mathcal{A}) = R_\mathcal{A}$ by the convexity of $\mathcal{F}_f$.*

*Proof.* For the rest of the proof, we use the notation $v$ for the unit vector $(\cos \rho, \sin \rho)$.

On the set of $n$-tuples of samples (namely $\mathbb{R}^n$), define the equivalence relation if two $n$-tuples are shifts of each other, that is, $\vec{x} = (x_1, \ldots, x_n)$ is in the same class as $\vec{y} = (y_1, \ldots, y_n)$ if and only if $y_1 - x_1 = \ldots = y_n - x_n$. Given a tuple $\vec{x}$, we use the notation $[\vec{x}]$ to denote the corresponding equivalence class. We will also abuse notation and use $[\vec{x}]$ to denote a unique representative chosen for that equivalence class, and use $\vec{x} - [\vec{x}]$ to denote the *scalar* shift between $\vec{x}$ and the representative $[\vec{x}]$ of its equivalence class.

Then, for an equivariant algorithm $\mathcal{A}$ whose output is denoted by $[\mu_\ell, \mu_r]$, we can write $R_\mathcal{A}$ as

$$R_\mathcal{A} = \mathop{\mathbb{E}}_{[\vec{x}]} \left[ \mathop{\mathbb{E}}_{\vec{x}|[\vec{x}]} \left[ \mathop{\mathbb{E}}_{\mathcal{A}|\vec{x}} [L(\mu_r - \mu_\ell)] \right] \right]$$

meaning that there are three levels of randomness: first draw an equivalence class $[\vec{x}]$ (according to the distribution $f = f^{(0)}$), then draw a random shift according to the conditional distribution $\vec{x} \mid [\vec{x}]$, then finally, the algorithm $\mathcal{A}$ can itself be random conditioned on its input $x$.

Similarly, we can write $\delta_\mathcal{A}$ as

$$\delta_\mathcal{A} = \mathop{\mathbb{E}}_{[\vec{x}]} \left[ \mathop{\mathbb{E}}_{\vec{x}|[\vec{x}]} \left[ \mathop{\mathbb{E}}_{\mathcal{A}|\vec{x}} \left[ \mathbb{1}_{0 \notin [\mu_\ell, \mu_r]} \right] \right] \right]$$

Given that the goal of Algorithm 3 is to optimize $\vec{v} \cdot (R_\mathcal{A}, \delta_\mathcal{A})$, linearity implies that it suffices to optimize, for each equivalence class $[\vec{x}]$, the inner product

$$\vec{v} \cdot \left( \mathop{\mathbb{E}}_{\vec{x}|[\vec{x}]} \left[ \mathop{\mathbb{E}}_{\mathcal{A}|\vec{x}} [L(\mu_r - \mu_\ell)] \right], \mathop{\mathbb{E}}_{\vec{x}|[\vec{x}]} \left[ \mathop{\mathbb{E}}_{\mathcal{A}|\vec{x}} \left[ \mathbb{1}_{0 \notin [\mu_\ell, \mu_r]} \right] \right] \right)$$

To see that Algorithm 3 optimizes the above inner product among all equivariant algorithms, first observe that by linearity, it suffices to consider deterministic algorithms $\mathcal{A}$ which output a deterministic interval on a given input $n$-tuple of samples.

Moreover, observe that the conditional distribution $\vec{x} \mid [\vec{x}]$ is, up to translation, equal to the normalized likelihood $l_{\vec{y}}(\theta)/\|l_{\vec{y}}\|_1$ for any $n$-tuple $\vec{y} \in [\vec{x}]$. This is because $\|l_{\vec{y}}\|_1$ is exactly the probability

(density) of the equivalence class $[\vec{x}]$. Therefore, in Step 4 of Algorithm 3, the probability $\delta_{[\tilde{\mu}_\ell, \tilde{\mu}_r]}$ is equal to the probability, under the conditional distribution $\vec{x} \mid [\vec{x}]$, that an equivariant algorithm whose output is $[\tilde{\mu}_\ell - (\vec{x} - [\vec{x}]), \tilde{\mu}_r - (\vec{x} - [\vec{x}])]$ under input $[\vec{x}]$ fails to output an interval that contains the parameter 0. That is, $\delta_{[\tilde{\mu}_\ell, \tilde{\mu}_r]} = \mathbb{E}_{\vec{x} \mid [\vec{x}]} \left[ \mathbb{1}_{0 \notin [\mu_\ell, \mu_r]} \right]$ where $[\mu_\ell, \mu_r]$ is the output of a deterministic equivariant algorithm that outputs $[\tilde{\mu}_\ell - (\vec{x} - [\vec{x}]), \tilde{\mu}_r - (\vec{x} - [\vec{x}])]$ under input $[\vec{x}]$.

To conclude, then, Algorithm 3 by construction chooses a deterministic interval to output so as to minimize the inner product

$$\vec{v} \cdot \left( \mathbb{E}_{\vec{x} \mid [\vec{x}]} \left[ \mathbb{E}_{\mathcal{A} \mid \vec{x}} \left[ L(\mu_r - \mu_\ell) \right] \right], \mathbb{E}_{\vec{x} \mid [\vec{x}]} \left[ \mathbb{E}_{\mathcal{A} \mid \vec{x}} \left[ \mathbb{1}_{0 \notin [\mu_\ell, \mu_r]} \right] \right] \right)$$

among all equivariant algorithms. The same reasoning also shows that the tiebreaking per input $n$-tuple sample $\vec{x}$ implies tiebreaking on $\delta_{\mathcal{A}}$ which is an expectation over the distribution of the $n$-tuple sample input. $\qquad \square$

Given Algorithm 3, we can now define the optimal estimator for a given confidence parameter $\delta$. which is by construction optimal given Theorem 4.3.

**Algorithm 4** (Minimax-optimal confidence-interval estimator). *Fix a distribution $f$. For an angle $\rho$, let $\delta_\ell(\rho)$ be the failure probability of Algorithm 3 with the flag smallestDelta set to true, denote this estimator by $\mathcal{A}_\ell(\rho)$, and $\delta_u(\rho)$ be the failure probability of Algorithm 3 with the flag set to false, similarly denoting this estimator by $\mathcal{A}_r(\rho)$. Given a failure probability $\delta^*$, by the supporting hyperplane theorem there exists an angle value $\rho$ (e.g. the arctangent of a sub-derivative of $R^*(\delta)$ at $\delta = \delta^*$) such that $\delta^* \in [\delta_\ell(\rho), \delta_u(\rho)]$. Let $\delta^* = \lambda\delta_\ell(\rho) + (1-\lambda)\delta_u(\rho)$. Then, define the optimal MSE estimator for failure probability $\delta$ to be the randomized algorithm which runs $\mathcal{A}_\ell(\rho)$ with probability $\lambda$ and runs $\mathcal{A}_r(\rho)$ otherwise. By definition, this estimator achieves $(R^*(\delta^*), \delta^*)$.*

The optimal slope angle $\rho$ can be approximately computed using binary search. We give the details in Appendix B.1.

## 4.2 Minimax optimality of Algorithm 4

In Section 4.1, we showed that Algorithm 4 is optimal among equivariant estimators. We now show that is in fact optimality among *all* estimators.

We first extend Definition 4.1 to cover also non-equivariant estimators.

**Definition 4.4.** *Given a (potentially) randomized $n$-sample estimator $\mathcal{A}$, whose output interval we denote by $[\mu_\ell, \mu_r]$, we say that $\mathcal{A}$ achieves its $(R_{\mathcal{A}}, \delta_{\mathcal{A}})$ pair if, for all parameters $\mu \in \mathbb{R}$, $\mathbb{E}_{x_1, \ldots, x_n \sim f^{(\mu)}}[L(\mu_r - \mu_\ell)] \leq R_{\mathcal{A}}$ and $\Pr_{x_1, \ldots, x_n \sim f^{(\mu)}}(\mu \notin [\mu_\ell, \mu_r]) \leq \delta_{\mathcal{A}}$.*

The key step in the proof of minimax optimality of Algorithm 4 is to show that the performance of any estimator can be well-approximated by an equivariant estimator.

**Proposition 4.5.** *Fix an arbitrary distribution $f$ and an arbitrarily small approximation parameter $b \in (0, \frac{1}{2})$. Suppose there is an estimator $\mathcal{A}$ (not necessarily equivariant) achieving $(R_{\mathcal{A}}, \delta_{\mathcal{A}})$, then there exists an equivariant estimator $\mathcal{A}'$ that achieves $((1 + b)R_{\mathcal{A}}, \delta_{\mathcal{A}} + 2b)$.*

We prove Proposition 4.5 in Appendix B. The main idea for constructing $\mathcal{A}'$ from $\mathcal{A}$ is that, on input an $n$-tuple of samples $(x_1, \ldots, x_n)$ whose sample median is at $s$, we 1) shift the samples by a random shift drawn from $\mathrm{Unif}[-t - s, t - s]$ for sufficiently large $t$, 2) call $\mathcal{A}$ on the shifted samples and 3) output the returned interval and subtract off the random shift we made to the samples. This estimator is equivariant by construction, and we relate its execution with high probability to applying $\mathcal{A}$ to samples drawn according to the prior $\mathrm{Unif}[-t, t]$. This allows us to bound the performance guarantees of $\mathcal{A}'$ by those of $\mathcal{A}$.

With Proposition 4.5, we can then show that Algorithm 4 is in fact minimax optimal among all estimators, not just equivariant ones.

**Theorem 4.6.** *Fix a distribution $f$. Recall the definition of $R^*(\delta)$ from Fact 4.2, which is the infimum of all achievable $(R_{\mathcal{A}}, \delta_{\mathcal{A}} = \delta)$ over all equivariant estimator $\mathcal{A}$. Now consider an arbitrary (potentially non-equivariant) estimator $\mathcal{A}'$. If $\mathcal{A}'$ achieves $(R_{\mathcal{A}'}, \delta_{\mathcal{A}'})$, then $R_{\mathcal{A}'} \geq R^*(\delta_{\mathcal{A}'})$.*

*As a result, since $(R^*(\delta^*), \delta^*)$ is achievable by Theorem 4.3 and Algorithm 4 for any $\delta^* > 0$, we have that Algorithm 4 is in fact minimax optimal among all estimators.*

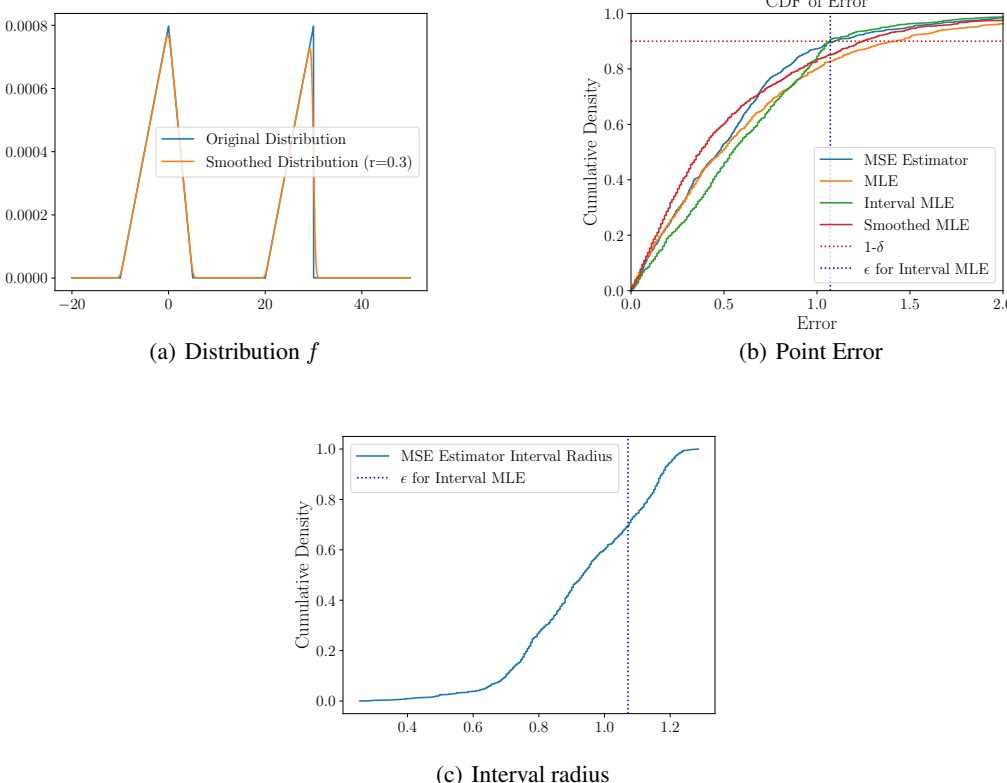

(a) Distribution $f$

(b) Point Error

(c) Interval radius

Figure 3: Example distribution and CDF of confidence interval widths

The proof is in Appendix B.

## 5 Experimental results

We compare various location estimation methods on synthetic data from a fairly simple, but irregular, piecewise linear distribution (Figure 3(a)). We set $n = 10$ and aim for $90\%$ confidence intervals. In Figure 3(b), we plot the CDF of the point error produced by the MLE, the 0.3-smoothed MLE, and our two algorithms (Algorithm 1 and Algorithm 4). We find that our algorithms get 25% smaller error than the MLE at the target 90% error rate. The smoothed MLE and Algorithm 4 error distributions both dominate the MLE, with Algorithm 4 having better performance close to its target and smoothed MLE better further away. Algorithm 1 does 2% better than Algorithm 4 at the target 90%, but worse over most of the range.

In Figure 3(c), we compare the confidence intervals produced by our algorithms. Algorithm 1 uses a fixed interval radius regardless of the samples it sees, while Algorithm 4 has a distribution over interval widths that is usually smaller but occasionally bigger.

## Acknowledgements

We thank the anonymous reviewers for insightful comments and suggestions on this work. Shivam Gupta and Eric Price are supported by NSF awards CCF-2008868, CCF-1751040 (CAREER), and the NSF AI Institute for Foundations of Machine Learning (IFML). Jasper C.H. Lee is supported in part by the generous funding of a Croucher Fellowship for Postdoctoral Research, NSF award DMS-2023239, NSF Medium Award CCF-2107079 and NSF AiTF Award CCF-2006206. Paul Valiant is supported by NSF award CCF-2127806.

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

# A  Missing details from Section 3

**Fact 3.2.** *Given an $n$-sample estimator $\hat{\mu}$, suppose there exists a sequence of priors $\{\pi_i\}$ such that the worst-case risk of $\hat{\mu}$ is equal to the limit of optimal Bayes risk of $\pi_i$, namely*

$$\sup_{\mu \in \mathbb{R}} R(\mu, \hat{\mu}) = \lim_{i \to \infty} \inf_{\hat{\mu}'} R(\pi_i, \hat{\mu}')$$

*Then $\hat{\mu}$ is minimax optimal.*

*Proof.* Consider any other estimator $\tilde{\theta}$.
Then

$$\sup_{\theta} R(\theta, \tilde{\theta}) \geq R(\pi_i, \tilde{\theta}) \geq \inf_{\hat{\theta}'} R(\pi_i, \hat{\theta}')$$

for all $\pi_i$, meaning that

$$\sup_{\theta} R(\theta, \tilde{\theta}) \geq \lim_{i \to \infty} \inf_{\hat{\theta}'} R(\pi_i, \hat{\theta}') = \sup_{\theta} R(\theta, \hat{\theta})$$

$\square$

## A.1  Binary search for approximately computing the optimal point estimator

In this section, we address the (approximate) computation of the optimal $\epsilon^*$ when given a desired failure probability $\delta^*$, using binary search. We explicitly give the (straightforward) binary search below in Algorithm 5, which accounts for the error in computing $\delta(\epsilon)$ for a given value of $\epsilon$. We also state the guarantees of Algorithm 5 here in Theorem A.1 and prove the theorem in Appendix A.

---

**Algorithm 5** Approximately computing $\epsilon^*$ from $\delta^*$ using binary search

---

Assume: Oracle for estimating $\delta(\epsilon)$ to additive error $\xi$ for any given $\epsilon$. Call this oracle $\hat{\delta}(\epsilon)$.
Input: Distribution $f = f^{(0)}$, desired failure probablity $\delta^*$

1. $\epsilon_{\max} \leftarrow 1$
2. While $\hat{\delta}(\epsilon_{\max}) + \xi > \delta^*$, double $\epsilon_{\max}$
3. $\epsilon_{\min} \leftarrow 0$
4. While $\epsilon_{\max} - \epsilon_{\min} > \eta$
   (a) $\epsilon_{\mathrm{mid}} \leftarrow (\epsilon_{\max} + \epsilon_{\min})/2$
   (b) If $\hat{\delta}(\epsilon_{\mathrm{mid}}) + \xi < \delta^*$, then $\epsilon_{\max} \leftarrow \epsilon_{\mathrm{mid}}$
   (c) Otherwise, $\epsilon_{\min} \leftarrow \epsilon_{\mathrm{mid}}$
5. Return $\epsilon_{\max}$

---

**Theorem A.1.** *Fix a distribution shape $f$, which implicitly defines a function $\epsilon^*(\delta)$ mapping a failure probability $\delta$ to an optimal estimation error $\epsilon^*(\delta)$. Algorithm 5, on input a failure probability $\delta^*$ and the distribution shape $f$, takes $\mathrm{poly}(\frac{1}{\xi}, |\log \epsilon^*(\delta^* - \xi)|, \log \frac{1}{\tau}, \log \frac{\epsilon^*(\delta^* - \xi)}{\eta})$ many executions of $\mathcal{A}_\infty$ and returns $\tilde{\epsilon} \in [\epsilon^*(\delta^*), \epsilon^*(\delta^* - \xi) + \eta]$ with probability $1 - \tau$.*

*Proof.* For now we analyze Algorithm 5 assuming the oracle. We will discuss the time complexity of the oracle and success probability of the whole algorithm at the end of the proof.

Given the guarantee that $|\hat{\delta} - \delta| \leq \xi$, at the end of Step 2, we have $\epsilon_{\max} \geq \epsilon^*$.

Due to the one-sided test in Step 4(b), by induction, we have that $\epsilon_{\min} \leq \epsilon^*(\delta^* - \xi)$ throughout the iterations of Step 4. Since we return $\epsilon_{\max}$ in Step 5, when $\epsilon_{\max} - \epsilon_{\min} \leq \eta$, we can guarantee that the returned value of $\epsilon$ is at most $\epsilon^*(\delta^* - \xi) + \eta$.

The above analysis assumes the success of every call to the oracle $\hat{\delta}$. Observe that $\delta$ is a failure probability, namely that it is the expectation of a Bernoulli coin. We can estimate $\delta$ via running $\mathcal{A}_\infty$

using phantom samples we generate from $f^0$. Via the standard Hoeffding bound, to yield an estimate of $\delta(\epsilon)$ to within additive error $\xi$ with probability $1 - \tau$, we need $O(\frac{1}{\xi^2} \log \frac{1}{\tau})$ samples, namely that many independent runs of $\mathcal{A}_\infty$. Since we make many calls to the oracle throughout Algorithm 5, it suffices for us to upper bound the number of oracle calls, and multiplicatively adjust $\tau$ appropriately such that the total failure probability is $\tau$ by a union bound, across all the oracle calls.

For Step 2, we can upper bound the number of iterations by $\log \max(1, \epsilon^*(\delta^* - \xi))$, since we always terminate whenever $\epsilon_{\max} \geq \epsilon^*(\delta^* - \xi)$ by the guarantee that $\hat{\delta} \geq \delta - \xi$.

Step 4 is just a binary search, halving the interval $[\epsilon_{\min}, \epsilon_{\max}]$ every time, from size $\max(1, \epsilon^*(\delta^* - \xi))$ down to $\eta$, thus requiring $\log(\max(1, \epsilon^*(\delta^* - \xi))/\eta)$ many iterations. Note also that $\log(\max(1, \epsilon^*(\delta^* - \xi))/\eta) \leq |\log \epsilon^*(\delta^* - \xi)| + \log \frac{\epsilon^*(\delta^* - \xi)}{\eta}$.

The runtime claim of Theorem A.1 thus follows. $\qquad \square$

# B  Missing details from Section 4

**Proposition 4.5.** *Fix an arbitrary distribution $f$ and an arbitrarily small approximation parameter $b \in (0, \frac{1}{2})$. Suppose there is an estimator $\mathcal{A}$ (not necessarily equivariant) achieving $(R_\mathcal{A}, \delta_\mathcal{A})$, then there exists an equivariant estimator $\mathcal{A}'$ that achieves $((1 + b)R_\mathcal{A}, \delta_\mathcal{A} + 2b)$.*

*Proof.* Given the estimator $\mathcal{A}$, first construct an intermediate estimator $\tilde{\mathcal{A}}$ such that, if $\mathcal{A}$ outputs an interval of length at most $L^{-1}(R_\mathcal{A}/b)$, then $\tilde{\mathcal{A}}$ outputs an identical output, otherwise $\tilde{\mathcal{A}}$ outputs an arbitrary interval of length at most $L^{-1}(R_\mathcal{A}/b)$. The claim is that $\tilde{\mathcal{A}}$ achieves $(R_\mathcal{A}, \delta_\mathcal{A} + b)$. It is immediate that $\tilde{\mathcal{A}}$ achieves mean loss at most $R_\mathcal{A}$ for any parameter $\mu$, by construction. As for its failure probability, for an arbitrary parameter $\mu$, the probability that $\mathcal{A}$ outputs an interval $[\mu_\ell, \mu_r]$ of length $> L^{-1}(R_\mathcal{A}/b)$ is at most $b$, since the mean loss is at most $R_\mathcal{A}$ for parameter $\mu$. Thus, the failure probability of $\tilde{\mathcal{A}}$ on parameter $\mu$ is at most $\delta_\mathcal{A} + b$, which holds true for any $\mu$.

We now construct the equivariant estimator $\mathcal{A}'$. On input $n$-tuple $\vec{x}$, let the sample median be $s$. Sample a number $w \sim \text{Unif}[-t - s, t - s]$ for some fixed parameter $t$ chosen later, where $t$ depends only on $b$ and the distribution $f$, and is independent of $\vec{x}$. Then, $\mathcal{A}'$ calls $\tilde{\mathcal{A}}$ on input $\vec{x} + (w, \ldots, w)$ and obtains an output interval $[\tilde{\mu}_\ell, \tilde{\mu}_r]$. Finally, $\mathcal{A}'$ outputs $[\tilde{\mu}_\ell - w, \tilde{\mu}_r - w]$.

Observe that the new estimator $\mathcal{A}'$ is by construction equivariant. It remains to choose the parameter $t$ as a function of $b$ (and the underlying distribution $f$). Consider drawing $n$ samples from $f = f^{(0)}$, then there must exist some distance $s_{\max}$ such that the sample median is within $s_{\max}$ of the origin with probability at least $1 - b^2/2$. We will choose the above parameter $t$ as $t = s_{\max}/b^2$.

Now we need to upper bound $R_{\mathcal{A}'}$ and $\delta_{\mathcal{A}'}$ in terms of $R_{\tilde{\mathcal{A}}}$ and $\delta_{\tilde{\mathcal{A}}}$ respectively. Specifically, we claim that there is a $1 - b^2$ probability coupling between the execution of $\mathcal{A}'$ on parameter $\mu = 0$ to the process of 1) drawing a parameter $\mu \sim \text{Unif}[-t, t]$, 2) drawing an $n$-tuple of samples $\vec{y}$ from $f^{(\mu)}$, 3) running $\tilde{\mathcal{A}}$ on $\vec{y}$ to get interval $[\tilde{\mu}_\ell, \tilde{\mu}_r]$ and outputting $[\tilde{\mu}_\ell - \mu, \tilde{\mu}_r - \mu]$. We note that the latter process achieves $(R_{\tilde{\mathcal{A}}}, \delta_{\tilde{\mathcal{A}}})$ with respect to parameter $\mu = 0$.

To show this, observe that with probability at least $1 - b^2/2$, drawing an $n$-tuple of samples $\vec{x}$ from $f = f^{(0)}$ will yield a sample median of $s \leq s_{\max}$, by definition of $s_{\max}$. Then, observe that since $t = s_{\max}/b^2$, the total variation distance between $\text{Unif}[-t - s, t - s]$ and $\text{Unif}[-t, t]$ is $s/2t \leq b^2/2$. Thus, with probability at least $1 - b^2$, the execution of $\mathcal{A}'$ on input samples drawn from $f^{(0)}$ is equal to a version of $\mathcal{A}'$ where the "shift" $w$ is instead always drawn from $\text{Unif}[-t, t]$. However, this variant of $\mathcal{A}'$ is exactly equivalent to the aforementioned process of 1) drawing a parameter $w \sim \text{Unif}[-t, t]$, 2) running $\tilde{\mathcal{A}}$ on samples drawn from $f^{(w)}$ and 3) outputting the interval after correcting for the $w$ shift.

Therefore, given this coupling, we conclude that $\delta_{\mathcal{A}'} \leq \delta_{\tilde{\mathcal{A}}} + b^2 \leq \delta_\mathcal{A} + b + b^2 \leq \delta_\mathcal{A} + 2b$ as $b < \frac{1}{2}$. As for $R_{\mathcal{A}'}$, since $\tilde{\mathcal{A}}$ always returns intervals whose loss is at most $R_\mathcal{A}/b$, in the probability $\leq b^2$ part of the coupling when the two processes behave differently, the contribution to the expected loss of $\mathcal{A}'$ is at most $b^2 \cdot R_\mathcal{A}/b = bR_\mathcal{A}$. Thus we can bound $R_{\mathcal{A}'}$ by $R_{\tilde{\mathcal{A}}} + bR_\mathcal{A} \leq (1 + b)R_\mathcal{A}$, as desired. $\qquad \square$

**Theorem 4.6.** *Fix a distribution $f$. Recall the definition of $R^*(\delta)$ from Fact 4.2, which is the infimum of all achievable $(R_{\mathcal{A}}, \delta_{\mathcal{A}} = \delta)$ over all equivariant estimator $\mathcal{A}$. Now consider an arbitrary (potentially non-equivariant) estimator $\mathcal{A}'$. If $\mathcal{A}'$ achieves $(R_{\mathcal{A}'}, \delta_{\mathcal{A}'})$, then $R_{\mathcal{A}'} \geq R^*(\delta_{\mathcal{A}'})$.*

*As a result, since $(R^*(\delta^*), \delta^*)$ is achievable by Theorem 4.3 and Algorithm 4 for any $\delta^* > 0$, we have that Algorithm 4 is in fact minimax optimal among all estimators.*

*Proof of Theorem 4.6.* Suppose for the sake of contradiction that $R_{\mathcal{A}'} < R^*(\delta_{\mathcal{A}'})$. By the supporting hyperplane theorem, there exists an angle $\rho \in [0, \pi/2]$ such that for all equivariant estimators $\mathcal{A}$, we have $\delta_{\mathcal{A}} \cos \rho + R_{\mathcal{A}} \sin \rho \geq \delta_{\mathcal{A}'} \cos \rho + R^*(\delta_{\mathcal{A}'}) \sin \rho$. However, by Proposition 4.5, for all $b \in (0, \frac{1}{2}]$, there is an equivariant estimator $\mathcal{A}_b$ that achieves $((1 + b)R_{\mathcal{A}'}, \delta_{\mathcal{A}'} + 2b)$. Consider a value of $b < \min(\frac{1}{2}, (R^*(\delta_{\mathcal{A}'}) - R_{\mathcal{A}'})/(2\cos \rho + R_{\mathcal{A}'} \sin \rho))$, noting that the upper bound is positive by our contradiction assumption. Then, it is straightforward to check that $\mathcal{A}_b$ violates the supporting hyperplane $\delta_{\mathcal{A}_b} \cos \rho + R_{\mathcal{A}_b} \sin \rho \geq \delta_{\mathcal{A}'} \cos \rho + R^*(\delta_{\mathcal{A}'}) \sin \rho$. Thus, it must be the case that $R_{\mathcal{A}'} \geq R^*(\delta_{\mathcal{A}'})$. $\square$

## B.1 Compute an optimal slope angle

In order to apply the construction of Algorithm 4, when given a distribution $f$ and a desired failure probability $\delta^*$, we need to compute an optimal slope angle value $\rho \in [0, \pi/2]$ to use for Algorithm 3. Since the failure probability is a monotonic function of $\rho$, the obvious strategy here is to again use binary search. Algorithm 6 returns an (approximately) optimal value of $\rho$ for Algorithm 3, as well as a probability for randomly setting the flag smallestDelta to true or false.

---

**Algorithm 6** Binary search for the optimal estimator through the slope angle $\rho$

---

Assume: Oracle for computing the failure probability of Algorithm 3 to additive error $\xi$ for any given angle $\rho$ and Boolean value for the smallestDelta flag. We will use $\hat{\delta}_\ell(\rho)$ to denote the oracle approximation for Algorithm 3 with smallestDelta set to true, and $\hat{\delta}_r(\rho)$ for smallestDelta set to false.

1. Input: distribution $f$, failure probability $\delta^*$

2. Set $\rho_{\min} \leftarrow 0$, $\rho_{\max} \leftarrow \pi/2$.

3. Repeat until $\hat{\delta}_\ell(\rho_{\min}) - \hat{\delta}_r(\rho_{\max}) \leq \sqrt{\xi} - 4\xi$:

   (a) Set $\rho_{\mathrm{mid}} \leftarrow (\rho_{\min} + \rho_{\max})/2$.

   (b) Use the oracle to evaluate the failure probabilities be $\hat{\delta}_\ell(\rho_{\mathrm{mid}})$ and $\hat{\delta}_r(\rho_{\mathrm{mid}})$ for Algorithm 3.

   (c) If $\delta^* < \hat{\delta}_\ell(\rho_{\mathrm{mid}}) + \xi$, $\rho_{\min} \leftarrow \rho_{\mathrm{mid}}$.

   (d) If $\delta^* > \hat{\delta}_r(\rho_{\mathrm{mid}}) + \xi$, $\rho_{\max} \leftarrow \rho_{\mathrm{mid}}$.

   (e) If $\delta^* \in [\hat{\delta}_\ell(\rho_{\mathrm{mid}}) + \xi, \hat{\delta}_r(\rho_{\mathrm{mid}}) + \xi]$

      i. If $\hat{\delta}_r(\rho_{\mathrm{mid}}) - \hat{\delta}_\ell(\rho_{\mathrm{mid}}) \leq \sqrt{\xi} - 4\xi$, terminate and return the algorithm that runs Algorithm 3 with $\rho = \rho_{\mathrm{mid}}$ and flag smallestDelta set to true always.

      ii. Otherwise, terminate and return the algorithm that runs Algorithm 3 with $\rho = \rho_{\mathrm{mid}}$ and flag smallestDelta set to true with probability $(1 + 3\sqrt{\xi})(\hat{\delta}_r(\rho_{\mathrm{mid}}) + \xi - \delta^*)/(\hat{\delta}_r(\rho_{\mathrm{mid}}) - \hat{\delta}_\ell(\rho_{\mathrm{mid}}))$.

4. Return the algorithm that runs Algorithm 3 with $\rho = \rho_{\max}$ and setting smallestDelta to be false always.

---

The following theorem captures the correctness of Algorithm 6.

**Theorem B.1.** *Fix an arbitrary distribution $f$ and failure probability $\delta^*$. If, for parameter $\xi$ smaller than some absolute constant, all the oracle calls in Algorithm 6 are correct, then, upon termination of Algorithm 6, the returned algorithm achieves a mean loss of $R_{\mathcal{A}} \in [R^*(\delta^*), R^*(\delta^* - O(\sqrt{\xi}))]$, and succeeds with probability at least $1 - \delta^*$.*

*Proof.* For this proof, denote $\delta_\ell(\rho)$ by the *actual* failure probability of Algorithm 3 using slope value $\rho$ and setting smallestDelta to true, and similarly for $\delta_r(\rho)$.

First, by induction, throughout the execution of the algorithm, we have that $\delta^* \in [\delta_r(\rho_{\max}), \delta_\ell(\rho_{\min}) + 2\xi]$. This can be checked straightforwardly by Steps 3(c,d) and the fact that $\hat{\delta}(\rho)$ is within additive $\xi$ error of $\delta(\rho)$ by assumption.

There are two ways the algorithm terminates, either in Step 4 or in Step 3(e).

We first analyze the termination condition in Step 4. Since, from the above paragraph, we have $\delta^* \in [\delta_r(\rho_{\max}), \delta_\ell(\rho_{\min}) + 2\xi]$. For the returned estimator from Step 4, its failure probability is by definition $\delta_r(\rho_{\max})$. In order to reach Step 4, it must be the case that $\hat{\delta}_\ell(\rho_{\min}) - \hat{\delta}_r(\rho_{\max}) \leq \sqrt{\xi} - 4\xi$, which implies $\delta_\ell(\rho_{\min}) - \delta_r(\rho_{\max}) \leq \sqrt{\xi}$. Therefore, we have $\delta_r(\rho_{\max}) \in [\delta^* - \sqrt{\xi}, \delta^*]$. By Fact 4.2, we conclude that $R_\mathcal{A} \in [R^*(\delta^*), R^*(\delta^* - \sqrt{\xi})]$.

Next, we analyze the two possible termination conditions in Step 3(e). We know that $\delta^* \in [\hat{\delta}_\ell(\rho_{\mathrm{mid}}) + \xi, \hat{\delta}_r(\rho_{\mathrm{mid}}) + \xi]$, which implies $\delta^* \in [\delta_\ell(\rho_{\mathrm{mid}}), \delta_r(\rho_{\mathrm{mid}}) + 2\xi]$.

If we terminate according to Step 3(e) i., then we know that $\hat{\delta}_r(\rho_{\mathrm{mid}}) - \hat{\delta}_\ell(\rho_{\mathrm{mid}}) \leq \sqrt{\xi} - 4\xi$. This implies $\delta_r(\rho_{\mathrm{mid}}) - \delta_\ell(\rho_{\mathrm{mid}}) \leq \sqrt{\xi} - 2\xi$. Combining with the last paragraph, we get $\delta_\ell(\rho_{\mathrm{mid}}) \in [\delta^* - \sqrt{\xi}, \delta^*]$. Also note that the returned estimator has failure probability $\delta_\ell(\rho_{\mathrm{mid}})$ by definition. Again using Fact 4.2, we conclude that the returned estimator satisfies $R_\mathcal{A} \in [R^*(\delta^*), R^*(\delta^* - \sqrt{\xi})]$.

Lastly, we check the case if we terminate according to Step 3(e) ii. The failure probability of the returned algorithm is

$$\frac{(1 + 3\sqrt{\xi})(\hat{\delta}_r(\rho_{\mathrm{mid}}) + \xi - \delta^*)}{\hat{\delta}_r(\rho_{\mathrm{mid}}) - \hat{\delta}_\ell(\rho_{\mathrm{mid}})} \delta_\ell(\rho_{\mathrm{mid}}) + \left(1 - \frac{(1 + 3\sqrt{\xi})(\hat{\delta}_r(\rho_{\mathrm{mid}}) + \xi - \delta^*)}{\hat{\delta}_r(\rho_{\mathrm{mid}}) - \hat{\delta}_\ell(\rho_{\mathrm{mid}})}\right) \delta_r(\rho_{\mathrm{mid}})$$

We will show that this failure probability is between $\delta^* - O(\sqrt{\xi})$ and $\delta^*$, which (again, by Fact 4.2) lets us conclude that $R_\mathcal{A} \in [R^*(\delta^*), R^*(\delta^* - O(\sqrt{\xi}))]$.

To show that the failure probability is at most $\delta^*$, observe that $\hat{\delta}_r(\rho_{\mathrm{mid}}) - \hat{\delta}_\ell(\rho_{\mathrm{mid}}) \leq \delta_r(\rho_{\mathrm{mid}}) - \delta_\ell(\rho_{\mathrm{mid}}) + 2\xi \leq (1 + 3\sqrt{\xi})(\delta_r(\rho_{\mathrm{mid}}) - \delta_\ell(\rho_{\mathrm{mid}}))$ since $\hat{\delta}_r(\rho_{\mathrm{mid}}) - \hat{\delta}_\ell(\rho_{\mathrm{mid}}) \geq \sqrt{\xi} - 4\xi$ by the condition in Step 3(e) ii. Furthermore, $\hat{\delta}_r(\rho_{\mathrm{mid}}) + \xi - \delta^* \geq \delta_r(\rho_{\mathrm{mid}}) - \delta^*$. Thus,

$$\frac{(1 + 3\sqrt{\xi})(\hat{\delta}_r(\rho_{\mathrm{mid}}) + \xi - \delta^*)}{\hat{\delta}_r(\rho_{\mathrm{mid}}) - \hat{\delta}_\ell(\rho_{\mathrm{mid}})} \geq \frac{\delta_r(\rho_{\mathrm{mid}}) - \delta^*}{\delta_r(\rho_{\mathrm{mid}}) - \delta_\ell(\rho_{\mathrm{mid}})}$$

implying that the failure probability is at most $\delta^*$.

Finally we need to show that the failure probability is at least $\delta^* - O(\sqrt{\xi})$. To see this, we upper bound

$$\frac{(1 + 3\sqrt{\xi})(\hat{\delta}_r(\rho_{\mathrm{mid}}) + \xi - \delta^*)}{\hat{\delta}_r(\rho_{\mathrm{mid}}) - \hat{\delta}_\ell(\rho_{\mathrm{mid}})}$$
$$\leq \frac{(1 + 3\sqrt{\xi})(\delta_r(\rho_{\mathrm{mid}}) - \delta^* + 2\xi)}{\delta_r(\rho_{\mathrm{mid}}) - \delta_\ell(\rho_{\mathrm{mid}}) - 2\xi}$$
$$\leq \frac{(1 + O(\sqrt{\xi}))(\delta_r(\rho_{\mathrm{mid}}) - \delta^* + 2\xi)}{\delta_r(\rho_{\mathrm{mid}}) - \delta_\ell(\rho_{\mathrm{mid}})} \quad \text{(since } \hat{\delta}_r(\rho_{\mathrm{mid}}) - \hat{\delta}_\ell(\rho_{\mathrm{mid}}) \geq \sqrt{\xi} - 4\xi)$$

Therefore, the failure probability is lower bounded by

$$\frac{(1 + O(\sqrt{\xi}))(\delta_r(\rho_{\mathrm{mid}}) - \delta^* + 2\xi)}{\delta_r(\rho_{\mathrm{mid}}) - \delta_\ell(\rho_{\mathrm{mid}})} \delta_\ell(\rho_{\mathrm{mid}}) + \left(1 - \frac{(1 + O(\sqrt{\xi}))(\delta_r(\rho_{\mathrm{mid}}) - \delta^* + 2\xi)}{\delta_r(\rho_{\mathrm{mid}}) - \delta_\ell(\rho_{\mathrm{mid}})}\right) \delta_r(\rho_{\mathrm{mid}})$$

$$= \delta^* - O(\sqrt{\xi})(\delta_r(\rho_{\mathrm{mid}}) - \delta^* + 2\xi) - (1 + O(\sqrt{\xi}))2\xi$$

$$\geq \delta^* - O(\sqrt{\xi}) \quad \text{since } \xi \text{ is sufficiently small}$$

$\square$

Since the failure probability $\delta$ is not a Lipschitz function of the slope angle $\rho$, we cannot in theory bound the runtime of Algorithm 6. However, we can nonetheless implement the probability estimation oracle and ensure a total failure probability of at most $1 - \tau$. As in Section 3.2, in order

to estimate the failure probability of an algorithm to accuracy $\xi$ with probability $1 - \tau$, it suffices to use $O(\log \frac{1}{\tau}/\xi^2)$ executions of Algorithm 3 over simulated samples (say, from $f^{(0)}$). In order to make sure that all oracle calls are correct, with a total failure probability of at most $\tau$, we simply decrease the failure probability of each subsequent oracle call by a factor of 2. That is, the first call is allowed to fail with probability $\tau$, then the second call $\tau/2$ and so on, which sums to a total failure probability of strictly less than $\tau$.

We emphasize again that these executions of Algorithm 3 use only "simulated" samples that we generate ourselves from a known distribution with a known shift, and not "real" samples we get for the estimation task.

