# OpenReview forum: "Minimax-Optimal Location Estimation"
_NeurIPS.cc/2023/Conference — NeurIPS 2023 poster_

### Official Review · Reviewer_j3jn · 2023-06-19

**Soundness:** 3 good
**Presentation:** 3 good
**Contribution:** 3 good
**Rating:** 6
**Confidence:** 3

**Summary:**

This work revisits one of the most basic questions in parametric statistics: location estimation. Two location estimators are proposed and proved to be optimal under different criteria.

From the reviewer's point of view, overall this paper is well-written and easy to understand. The problem is well-motivated, and the mathematical derivation seems rigorous. I only have some minor comments.

1. is it possible to provide some motivation before presenting Algorithm 1 and 2？ So readers can have a better understanding of it.

2. Whether the theoretical results of classical MLE can be established under these different criteria？

3. I understand due to the page limit, only one pdf is considered. Maybe some classical pdf like Gaussian can also be considered to better compare MLE and the proposed methods.

4. Is it possible to put Figure 2 after Section 5. The current location is a little bit weird.

**Strengths:**

Please see summary.

**Weaknesses:**

Please see summary.

**Questions:**

Please see summary.

**Limitations:**

Please see summary.

---

> ### Author Rebuttal · Authors · 2023-08-09
>
> Thank you for your suggestions and comments.
>
> Q1: We will provide more motivation before Algorithm 1, as discussed in the overall response.
>
> Q2: There are prior works studying the finite-sample guarantees of MLE for point estimation (see [Spo11, Pin17, VdG00, Mia10]), but they tend to lose constants (at least) or make strong assumptions. We can augment the MLE into an interval estimate simply by using an interval of size equal to the estimation error guarantees in these other works, but naturally the guarantees are rather sub-optimal compared to what we show for Algorithm 3 (which is optimal).
>
> [Spo11] Vladimir Spokoiny. Parametric estimation. finite sample theory. The Annals of Statistics, 40(6):2877--2909, 2011.
>
> [Pin17] Iosif Pinelis. Optimal-order uniform and nonuniform bounds on the rate of convergence to normality for maximum likelihood estimators. Electronic Journal of Statistics, 11(1):1160--1179, 2017.
>
> [VdG00] Sara A Van de Geer. Empirical Processes in M-estimation, volume 6. Cambridge university press, 2000.
>
> [Mia10] Yu Miao. Concentration inequality of maximum likelihood estimator. Applied Mathematics Letters, 23(10):1305–1309, 2010.
>
> Q3:  A Gaussian pdf wouldn't be very interesting, because the empirical mean is optimal for every confidence level, and basically every method will output it.  (In the Bayesian interpretation, the posterior is Gaussian, which is symmetric.)  But we might show a Laplacian, depending on space.

---

> > ### Comment · Reviewer_j3jn · 2023-08-14
> > **Response**
> >
> > Thanks for the response. I will keep the score as it is.

---

### Official Review · Reviewer_b2QA · 2023-07-06

**Soundness:** 3 good
**Presentation:** 2 fair
**Contribution:** 3 good
**Rating:** 5
**Confidence:** 2

**Summary:**

This paper studies the problem of location estimation. We obtain n i.i.d. samples from a distribution f(x-u*) where f is known and u* is fixed but unknown. The goal is to estimate u* as accurately as possible. This is a generalization of Gaussian mean estimation though it's not as difficult of the general mean estimation because f is known. Note that for large n, maximum likelihood estimator gives asymptotically optimal performance, so the focus is on small n.

The authors present two algorithms for optimal "average case" and "worst case" behavior.
1) Minimize estimation error |u - u*| with failure probability \delta.
2) Minimize length of an interval [u_l, u_r] which contains u* with probability 1-\delta. This algorithm can be generalized to any loss function of the length of the interval, L(u_r - u_l).

**Strengths:**

Essentially optimal error guarantees for small n in both "worst case" and "average case" scenarios.

**Weaknesses:**

W1. Does not discuss running times.
W2. Fig 2b shows that the landscape of performance is different for smaller error (and success probabilities). Is this because of how epsilon is chosen, or is it inherent?

**Questions:**

See weaknesses

**Limitations:**

See above

---

> ### Author Rebuttal · Authors · 2023-08-09
>
> **Running times.** One challenge in expressing the running time is it depends on how the distribution is represented as an input.  If it's just given as a density discretized to $N$ bins, then Algorithm 1 can be done in $O(N \log N)$ time by an FFT convolution to compute the likelihoods.  Algorithm 2 just runs Algorithm 1 inside a binary search, so multiplies the time by log of the precision desired.
> Algorithm 3 takes $O(N^2)$ time, and Algorithm 4 adds another logarithmic factor from binary search.
>
> **Landscape of performance.** Our algorithms depend on the desired confidence level $1-\delta$, for which they optimize the max/average confidence interval size.  By contrast, the MLE/smoothed MLE do not depend on the desired confidence level. In Figure 2b, we asked our algorithms to optimize for 90\% coverage.
>
> They quite correctly outperform the competing methods at 90\% coverage, but there is a cost: worse performance in significantly different regimes (e.g., worse median error than smoothed MLE).
> This tradeoff is inherent to optimizing 90\% coverage.

---

> > ### Comment · Reviewer_b2QA · 2023-08-11
> > **Acknowledgement**
> >
> > Thanks for responding to the questions. I will keep my score at 5. This is a good paper but has room for improvement.
> > 1) Smoothed MLE works pretty well, even at the 90% coverage the authors optimize their algorithms for (error rate ~1.2 vs ~1.1) and it outperforms the proposed algorithms on most of the range (which the algorithm is not optimized for).
> > 2) Evaluations could be more comprehensive on a wide range of real and synthetic datasets with larger "n".
> >
> > In the discussion of runtime, what would happen if one only has access to a sampling oracle (rather than an explicitly discretized density function)?

---

### Official Review · Reviewer_ng2F · 2023-07-08

**Soundness:** 3 good
**Presentation:** 3 good
**Contribution:** 3 good
**Rating:** 6
**Confidence:** 3

**Summary:**


This paper considers the problem of estimating an unknown shift given samples from a known density. The authors provide two estimators for this problem” a point estimator and an output interval, and they demonstrate these estimators are minimax optimal in slightly different ways. They provide an empirical evaluation of their estimators as well.

**Strengths:**

Overall I found the paper well written and the authors did a good job motivating their constructions and the results they provided. Section 1.1 provided a solid summary that made getting through the rest of the paper easier.

**Weaknesses:**


There were several places where I believe the paper could be improved or I struggled with the paper.

Major comments:
1. Significance: The authors mention that this question is a fundamental one in parametric statistics, but I totally failed to understand why. I understand it’s a theory paper - but regardless, Some more motivation would help the reader understand why having better methods is important.
2. More motivation around what algorithm 1 is accomplishing - perhaps on an example, similar to the mixture discussed in the introduction would help. I didn’t really understand why the algorithm was natural without reading the proof.
3. Algorithm 1/3 are very much designed with the goal of being optimal with the loss function  defined in 3.1 How does that objective compare to the guarantee of previous works (i.e. the smoothed mle). In particular, those results seem a bit stronger since they guarantee convergence inference on mu through asymptotic convergence in distribution. Can the authors remark on whether a Bayes optimal estimator will have similarly strong properties?
4. Following up on this, it would have been nice if the experimental section had been more flushed out. It was unclear what the take away was and only one example was provided. Are there regimes where the estimators of these papers perform better/worse than other estimators? Where should I expect the estimator to perform well?

Minor Comments:
1. I found the proof of Theorem 1.2 reasonably straightforward to follow however the proof of 4.3 was more challenging. I think having a picture to demonstrate the underlying geometry of the 2d Pareto front would have helped significantly. Also, the discussion in lines 284-290 was quite tricky - I still don’t fully understand what is the conditional distribution being discussed.
2. Note that the risk function R being used in section 3.1 is overloaded in notation. You have R(hat mu, mu) vs R (pi, mu). This is not a huge problem, but I did find it a bit confusing.


**Questions:**

See above

---

> ### Author Rebuttal · Authors · 2023-08-09
>
> Thank you for your questions and suggestions.
>
> (1) The goal of our (long-term) project is to build a finite-sample minimax theory of parametric statistics.  Classical theory is asymptotic, without finite-sample guarantees; location estimation is a simple setting to start building this theory, so we start there and already find interesting behavior.
>
> (2) See the general response: Algorithm 1 is essentially the Bayes optimal algorithm (fixing estimation error $\epsilon$ and optimizing over failure probability $\delta$) for $\theta$ drawn from a uniform prior over the real line.  (This isn't an actual prior, but it's the limiting behavior).
>
> (3) Under smoothness assumptions, our estimator will also be asymptotically normal: it's an interval version of MLE, where the interval length goes to 0 as $n \to 0$.  Without smoothness assumptions, we're not sure -- but as far as we know, it's not even known if the standard MLE is asymptotically normal under zero assumptions.
>
> (To be clear, the above is discussing asymptotic normality of $\sqrt{n}$ times the error.  For distributions with infinite Fisher information, like the uniform distribution, the error rate is $\ll 1/\sqrt{n}$.  The error of our estimator then does *not* look normal at its intrinsic scale, but when scaled by $\sqrt{n}$ it is essentially zero and hence $N(0, 0)$.)
>
> (4) The minimax estimators are, by construction, optimal. So Algorithm 1/2 (Interval MLE) will never perform worse than any other estimator, at the specified confidence level; but it may be significantly worse for other confidence levels.
> The takeaway of the experiment is to confirm this, and to compare the overall behavior of Interval MLE vs the minimax-optimal interval estimator (Algorithm 3/4, MSE Estimator). The experiment shows that, while the MSE estimator performs slightly worse at the specified (90\%) confidence than Interval MLE, its average case behavior is (much) better.
>
>
> Minor comment 1: We included a figure of the 2d Pareto front in the rebuttal pdf that we'll include in the final paper. Do you have any comments on it?

---

> > ### Comment · Reviewer_ng2F · 2023-08-15
> >
> > Thanks for the Pareto front picture - it is indeed helpful.  I also appreciate the responses to my queries. I will raise my score accordingly.

---

### Official Review · Reviewer_cZWD · 2023-07-13

**Soundness:** 3 good
**Presentation:** 4 excellent
**Contribution:** 4 excellent
**Rating:** 8
**Confidence:** 3

**Summary:**

This paper is about location estimation from sampling. That is, given n samples from a known probability distribution f, which is shifted by an unknown value $\mu$, the goal is to estimate that shift. Gaussian mean estimation is a special case of this parametric estimation type of problem. The paper describes two minimax optimal algorithms with respect to the failure probability and the accuracy, which output a confidence interval for $\mu$. The first algorithm achieves this in a worst-case sense. That is, it is trying to output as small confidence interval as possible in the worst case instance. However, the second one is minimax optimal with respect to the mean loss, which is the expected value of a loss function which is increasing with respect to the length of the confidence interval.
The authors motivate the use of mean loss by constructing a mixture of two gaussians consisting of one with high mass and high variance and a second one with very small mass and small variance, so that the existence of samples from the second Gaussian is rare, but provides much more information about the shift $\mu$ when it occurs.


**Strengths:**

The paper gives optimal results on a very fundamental problem of location estimation in the finite sample setting. The paper is also quite well written, while highly non-trivial techniques are used.

**Weaknesses:**

N/A

**Questions:**

N/A

---

> ### Author Rebuttal · Authors · 2023-08-09
>
> Thank you for your positive appreciation of our results.

---

### Author Rebuttal · Authors · 2023-08-09

We thank the reviewers for the positive reviews of our work. In particular, it is encouraging that the reviewers find our work 1) tackling an important problem (reviewers cZWD, j3jn), with 2) strong motivations (reviewers ng2F, j3jn) as well as 3) using highly non-trivial techniques (reviewer cZWD) to give optimal error guarantees.
We will respond to the individual reviews.
In this overall response, we only wish to address the question of technically motivating the construction of Algorithm 1, since it was raised by both reviewers ng2F and j3jn.

Recall that Algorithm 1 is the point estimator which, fixing an estimation accuracy $\epsilon$, attains the minimax-optimal failure probability.
The key intuition is that, since we don't know what the true parameter $\theta$ is, we might as well start by constructing an estimator that is Bayes optimal with respect to the uniform prior over the real line $\theta \sim \mathrm{Unif}(\mathbb{R})$.
But, there is no such thing as a uniform prior over the real line.
So instead, Algorithm 1 is constructed to be Bayes optimal with respect to the prior $\theta \sim \mathrm{Unif}[-t,t]$, and we take the limit of $t \to \infty$ to get a translation-equivariant estimator, whose risk (failure probability) is independent of the underlying parameter $\theta$.
We then use standard tools to relate Bayes optimality to minimax optimality, under such translation equivariance.

---

### Decision · Program_Chairs · 2023-09-21

**Decision:**

Accept (poster)

**Comment:**

The paper focuses on location estimation from samples, aiming to estimate an unknown shift $\mu$ in a probability distribution $f$ based on n samples. It covers Gaussian mean estimation as a special case. Two minimax optimal algorithms are introduced: one for minimizing failure probability by generating tight confidence intervals, and another for minimizing mean loss by considering the expected value of a length-increasing loss function. The latter is motivated by a mixture of two Gaussians, emphasizing the importance of rare samples from a low-mass, low-variance Gaussian for accurate estimation of $\mu$.